# Magnetometer-Guided Sentinel Lymph Node Dissection in Prostate Cancer: Rate of Lymph Node Involvement Compared with Radioisotope Marking

**DOI:** 10.3390/cancers13225821

**Published:** 2021-11-20

**Authors:** Svenja Engels, Bianca Michalik, Luca-Marie Meyer, Lena Nemitz, Friedhelm Wawroschek, Alexander Winter

**Affiliations:** University Hospital for Urology, Klinikum Oldenburg, Department of Human Medicine, School of Medicine and Health Sciences, Carl von Ossietzky University Oldenburg, 26122 Oldenburg, Germany; engels.svenja@klinikum-oldenburg.de (S.E.); michalik.bianca@klinikum-oldenburg.de (B.M.); luca-marie.meyer@uol.de (L.-M.M.); nemitz.lena@klinikum-oldenburg.de (L.N.); wawroschek.friedhelm@klinikum-oldenburg.de (F.W.)

**Keywords:** prostate cancer, sentinel lymph node, lymphadenectomy, metastases, superparamagnetic iron oxide nanoparticles, radioisotopes

## Abstract

**Simple Summary:**

Pelvic lymph node dissection is recommended in prostate cancer according to the patients’ individual risk for nodal metastases. Targeted removal of sentinel lymph nodes increases the number of detected lymph node metastases in patients with prostate cancer. We previously established magnetometer-guided sentinel lymph node dissection in patients with prostate cancer to overcome logistical and technical disadvantages associated with the standard radioisotope-guided technique. This retrospective study compared the magnetometer-guided and standard techniques in terms of their ability to detect lymph node metastases. Using the magnetometer-guided technique, more sentinel lymph nodes were detected per patient. The detected rates of lymph node involvement matched the predictions in both techniques equally well. Our findings confirm the reliability of magnetometer-guided sentinel lymph node dissection and highlight the importance of the sentinel technique for detecting lymph node metastases in prostate cancer.

**Abstract:**

Sentinel pelvic lymph node dissection (sPLND) enables the targeted removal of lymph nodes (LNs) bearing the highest metastasis risk. In prostate cancer (PCa), sPLND alone or combined with extended PLND (ePLND) reveals more LN metastases along with detecting sentinel LNs (SLNs) outside the conventional ePLND template. To overcome the disadvantages of radioisotope-guided sPLND in PCa treatment, magnetometer-guided sPLND applying superparamagnetic iron oxide nanoparticles as a tracer was established. This retrospective study compared the nodal staging ability between magnetometer- and radioisotope-guided sPLNDs. We analyzed data of PCa patients undergoing radical prostatectomy and magnetometer- (848 patients, 2015–2021) or radioisotope-guided (2092 patients, 2006–2015) sPLND. To reduce heterogeneity among cohorts, we performed propensity score matching and compared data considering sentinel nomogram-based probabilities for LN involvement (LNI). Magnetometer- and radioisotope-guided sPLNDs had SLN detection rates of 98.12% and 98.09%, respectively; the former detected more SLNs per patient. The LNI rates matched nomogram-based predictions in both techniques equally well. Approximately 7% of LN metastases were detected outside the conventional ePLND template. Thus, we confirmed the reliability of magnetometer-guided sPLND in nodal staging, with results comparable with or better than radioisotope-guided sPLND. Our findings highlight the importance of the sentinel technique for detecting LN metastases in PCa.

## 1. Introduction

Pelvic lymph node dissection (PLND) is the most reliable technique for lymph node (LN) staging in clinically localized prostate cancer (PCa) [1]. LN status is a therapeutically crucial prognostic factor in PCa because the presence and extent of LN involvement (LNI) are related to an increased risk of systemic dissemination and progression of the disease [2,3,4]. Moreover, PLND or resection of LN metastases has been indicated to have therapeutic benefits, particularly in patients with minimal LNI [1,5,6,7,8].

The detection of LNI directly correlates with the number of dissected LNs as well as with the anatomical limits of PLND [9,10]. The European Association of Urology (EAU) guidelines, therefore, recommend an extended PLND (ePLND) for LN staging in patients with >5% risk of LNI as diagnosed by systematic random biopsy [11,12] or in those with >7% risk of LNI as diagnosed by multiparametric magnetic resonance imaging (MRI) and MRI-targeted biopsy [12,13].

The complication rate, however, also increases alongside the increase in the number of LNs removed [14,15,16]. Therefore, Wawroschek et al. [17] adopted the techniques and concepts of radioisotope-guided sentinel LN (SLN) identification from other tumor entities for use in PCa, and this has subsequently been independently confirmed in several studies [18,19]. Sentinel PLND (sPLND) enables the targeted removal of clinically negative LNs, which bear a high probability of containing metastases [20] because SLNs are the first lymphatic drainage stations of their primary organs or the respective tumor [21]. In PCa, sPLND alone or in combination with ePLND increases the number of detected LN metastases [19,22,23]. Moreover, during sPLND, SLNs occurring outside the conventional ePLND template can be removed [22,23,24]. Thus, sPLND provides additional diagnostic value by adjusting the degree and anatomical extent of PLND to the patient’s individual lymphatic drainage situation [19,22,23,25].

Unfortunately, the use of radioisotope tracers for SLN marking is associated with several technical and logistical disadvantages [26,27]. For example, the practicality of the radioisotope-guided sPLND technique depends on the accessibility to radioisotope tracers and nuclear medicine facilities. Thus, this technique is used only in more developed countries or hospitals with access to such technology. Furthermore, this technique exposes patients and surgical staff to radiation, an aspect that is strongly controlled by legislation. In patients with breast cancer, superparamagnetic iron oxide nanoparticles (SPIONs) have been successfully developed as an equivalent, easy-to-use, and radiation-free alternative for SLN marking and intraoperative detection [26,28]. Our group subsequently adopted this technique of intraoperative magnetic detection of SLNs for use in patients with PCa; we use a system that comprises a magnetic tracer and a handheld magnetometer [23,24,27,29].

This retrospective study primarily aimed to compare magnetometer- and radioisotope-guided sPLND techniques in terms of their ability to detect LN metastases in patients with PCa. The secondary aim was to evaluate the anatomical distribution of dissected SLNs and detected LN metastases. We analyzed two large data sets from patients with PCa who underwent radical prostatectomy in combination with either of the two sPLND techniques at a tertiary referral hospital and performed a matched-pair analysis. The rates of LNI were compared between the two techniques while considering the patients’ individual probabilities for LNI as inferred from our sentinel nomogram [30].

## 2. Materials and Methods

### 2.1. Patient Populations

This retrospective study considered two cohorts of patients with PCa consecutively documented in the database of the University Hospital for Urology Oldenburg. We initially enrolled 2186 patients with PCa who underwent open retropubic radical prostatectomy combined with radioisotope-guided sPLND between January 2006 and February 2015. Then, we excluded the data of 11 patients who received only one-sided sPLND and of one patient in whom the time between tracer injection and surgery extended the manufacturer-guaranteed tracer detectability period. Furthermore, 45 patients who underwent hormonal treatment and 34 who underwent transurethral prostate surgeries before prostatectomy were also excluded. LN dissection data were incomplete in three additional patients. The final sample for the analysis of radioisotope-guided sPLND included 2092 patients.

Initially, we included 881 patients with PCa who underwent open retropubic radical prostatectomy combined with magnetometer-guided sPLND between February 2015 and May 2021. Then, we excluded the data of seven patients who received only one-sided sPLND, three patients with limited tracer detectability because of metal implants, and two patients in whom the time between tracer injection and surgery extended the manufacturer-guaranteed detectability period. Furthermore, we also excluded 16 patients who underwent hormonal treatment and 4 who underwent transurethral prostate surgeries prior to prostatectomy. The LN dissection data were incomplete in one additional patient. The final sample for the analysis of magnetometer-guided sPLND included 848 patients.

All patients were informed verbally and in writing about the open retropubic radical prostatectomy and sPLND; all signed a consent form before surgery.

### 2.2. sPLND Technique and Histopathological Examination

All patients were administered with transrectal tracer injection, either 99mTechnetium nanocolloid (160 MBq Nanocoll^®^, Nycomed Amersham Sorin, Milan, Italy) or superparamagnetic iron oxide nanoparticles (02/2015–01/2019 Sienna+^®^, Sysmex Europe GmbH, Norderstedt, Germany; 01/2019–05/2021 Magtrace^®^, Sysmex Europe GmbH, Norderstedt, Germany, into the prostate under ultrasonic guidance a day before surgery. SPLND was performed as described by Wawroschek et al. [31] and Winter et al. [32] for radioisotope-guided surgery and as described by Winter et al. [23] for magnetometer-guided surgery. All SLNs detected by a gamma probe (C-Trak System, Care Wise, Morgan Hill, CA, USA, or Crystal Probe SG04, Crystal Photonics GmbH, Berlin, Germany) or a magnetometer (Sentimag^®^, Sysmex Europe GmbH, Norderstedt, Germany) together with lymphatic fatty tissue directly adjoining or adhering to the identified SLNs were removed surgically. After sentinel-guided surgery, risk-adapted or if no SLN was detected at all, PLND was completed by ePLND using the anatomic template as described by Weingärtner et al. [33], which included all lymphatic fatty tissue along the external and internal iliac vessels and the obturator fossa as well as within the area dorsal to the obturator nerve, from the bifurcation of the common iliac artery (proximal limit) to the femoral canal (distal limit) and from the pelvic sidewall (lateral limit) to the perivesical fatty tissue (medial limit).

After surgery, LNs were cut into 3 mm transverse sections, routinely processed, and embedded into paraffin. Then, 4–5 µm sections were stained with hematoxylin-eosin (HE; Figure 1). In rare cases of inconclusive conventional histology, samples were immunohistochemically stained with AE1/AE3 pancytokeratin antibodies to check for (micro-)metastases.

### 2.3. Data Analyses

All data analyses were performed using R 4.1.0 [34]. For each sPLND technique, we calculated the median numbers of SLNs detected per patient, SLN detection rate (proportion of patients with detected SLNs), rate of LNI (proportion of patients with pathologic nodal stage 1; pN1), and the false-negative rate (proportion of LN-positive but SLN-negative cases). Numbers and proportions were compared statistically using the Wilcoxon rank-sum test and χ^2^ proportions test, respectively. Tests were repeated after 1:1 optimal pair matching according to clinical information (i.e., age, prostate-specific antigen, clinical tumor stage, Gleason score, and percentage of positive biopsy cores) based on the results of a propensity score analysis. A summary of the propensity score-adjusted data is presented in Table A1 of Appendix A. We estimated the probability of LNI for each patient according to our nomogram [30]. This nomogram predicts a patient’s individual probability for the presence of lymphogenic metastases based on clinical information such as prostate-specific antigen (PSA) value, clinical tumor stage, biopsy Gleason score, and percentage of positive biopsy cores [30]. In our data set, clinical tumor stage and/or biopsy data (biopsy Gleason score and/or the percentage of tumor-positive biopsy cores) were unavailable for 24 patients. To evaluate the diagnostic accuracy of each sPLND technique, we plotted the observed LNI rates against the nomogram-predicted probabilities of LNI in steps of approximately 3.33% (30 bins). Curves were smoothed, and 95% confidence intervals were calculated using the Loess method (local polynomial regression fitting). A χ^2^ goodness-of-fit test was computed for each technique to estimate the deviation of the curves from the ideal curve wherein observed, and predicted rates of LNI match perfectly. Moreover, the numbers of dissected SLNs and LN metastases were counted for each anatomical region of sPLND. The region data were unavailable for 22 SLNs detected during magnetometer-guided sPLND.

## 3. Results

In total, we dissected 12,331 LNs in 848 patients who underwent magnetometer-guided sPLND and 22,565 LNs in 2092 patients who underwent radioisotope-guided sPLND. Among these dissected LNs, 6478 and 12,981 were SLNs. Table 1 summarizes clinical and histopathological tumor characteristics as well as LN dissection data of the two patient cohorts. A summary of these data after propensity score matching can be found in Table A1. The SLN detection rates did not differ between the magnetometer- and radioisotope-guided sPLND techniques (98.11%, *n* = 832 patients vs. 98.18%, *n* = 2054 patients, respectively; Table 2). However, we detected significantly more SLNs per patient using magnetometer-guided sPLND (median = 7, IQR: 4–10) than that using radioisotope-guided sPLND (median = 6, IQR: 4–8; Table 2).

We found metastases in 621 LNs of 193 patients who underwent magnetometer-guided sPLND and in 1010 LNs of 396 patients who underwent radioisotope-guided sPLND (Table 1). Figure 1 shows a representative example of a HE staining of an SLN metastasis as revealed by the magnetic tracer. On the one hand, there were significantly more patients with LN positivity among those who underwent magnetometer-guided sPLND than among those who underwent radioisotope-guided sPLND (22.76% vs. 18.93%, respectively; Table 2), which leveled off after propensity score matching (Table 2). On the other hand, no difference was noted between the two techniques in the proportion of patients who had metastases only in non-SLNs (7.25%, *n* = 14 vs. 9.85%, *n* = 39, respectively; Table 2). Of these, in five patients who underwent magnetometer-guided sPLND and in 15 patients who underwent radioisotope-guided sPLND, respectively, no SLNs could be detected at all, and in two and four patients, respectively, macroscopically visible metastases were surgically removed without measuring tracer activity. Excluding these cases from the sample of only patients with non-SLN positivity, the resulting false-negative rates were 3.63% (*n* = 7) for magnetometer-guided sPLND and 5.05% (*n* = 20) for radioisotope-guided sPLND (Table 2).

**Figure 1 cancers-13-05821-f001:**
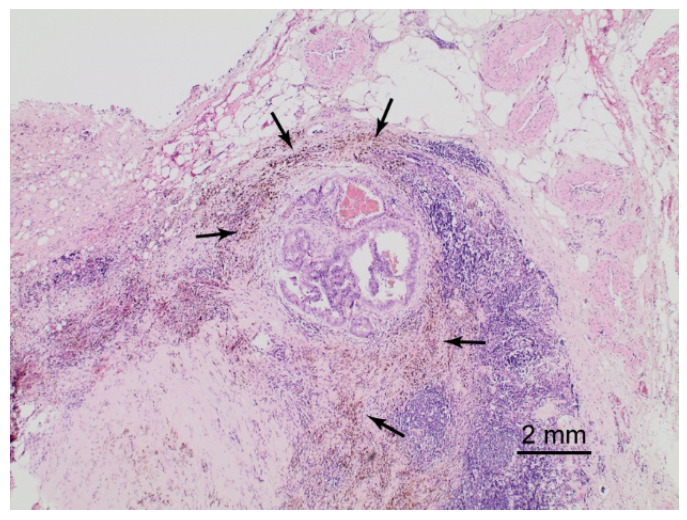
Hematoxylin-eosin staining of a magnetically traced sentinel lymph node. In the center of the image, a 4 mm metastasis of a Gleason 8 adenocarcinoma of the prostate is present. Note the brownish discoloration (arrows) of macrophages containing the magnetic tracer (superparamagnetic iron oxide nanoparticles). Total magnification 40×.

The observed proportions of patients with LN positivity did not differ from the proportions predicted by the nomogram in both magnetometer- (goodness-of-fit test: χ^2^ = 0.73, df = 29, *p* = 1) and radioisotope-guided (goodness-of-fit test: χ^2^ = 0.75, df = 29, *p* = 1; Figure 2) sPLND techniques.

Anatomic regions outside the standard template for ePLND accounted for 4.5% of SLNs detected by a magnetometer and 2.9% of SLNs detected by a radioprobe, respectively (Figure 3a). Accordingly, approximately 7.6% and 5.7% of LN metastases detected by magnetometer-guided and radioisotope-guided sPLND, respectively, occurred in LNs outside the standard ePLND template (Figure 3b).

## 4. Discussion

In this retrospective study of pelvic sentinel lymphadenectomy in patients with PCa, SLN detection rates were equally high for both magnetometer- and radioisotope-guided sPLND techniques; however, the former revealed a greater number of SLNs per patient. The accordance between the observed rates of LNI and nomogram-based predictions was equally high in both sPLND techniques. Both sPLND techniques revealed a considerable proportion of lymphogenic metastases outside the conventional ePLND template.

The SLN detection rates found in our study closely match those reported in other studies on sPLND in PCa [19,22,35]. Our observed rates of LNI were accordingly high [19,32,36] and even higher than expected from the ePLND data [37,38,39,40]. We resect a comparatively large number of SLN with both procedures. This can probably be explained by the fact that we do not perform dynamic imaging in clinical routine and remove all active LNs regardless of their activity level. Therefore, strictly speaking, our procedure could rather be called “lymphatic mapping”.

To predict the individual probability of LNI in patients with PCa, Winter et al. [30] developed a nomogram, which is based on clinical information and radioisotope-guided sPLND data. This nomogram was subsequently validated externally [41]. Our analyses revealed high accordance between nomogram-based predictions and LNI rates in both sPLND techniques. These results suggest that magnetometer-guided sPLND is a reliable and promising alternative that can be used in PCa to overcome the disadvantages of radioisotope marking [26,27], as observed in other tumor entities [26,28,42,43,44,45]. Furthermore, an improved flow of the magnetic tracer may lead to an increased number of detected SLNs in magnetometer-guided sPLND, as observed in the present study. The size of the magnetic tracer particles is approximately 60 nm, which is slightly smaller and more homogeneous than the radioisotope tracer [26]. The radioisotope tracer particles were 80 nm (95%), 80–100 nm (4%), and >100 nm (1%) [27]. Thus, the particle size might have influenced the drain of the two different tracers through the lymphatic pathways and/or trapping in the LNs, leading to more SLNs being detected using magnetometer-guided sPLND.

At our hospital, the magnetometer-guided sPLND has largely replaced radioisotope-guided sPLND, which is an obvious limitation of our retrospective study design as the two techniques were used in different study periods. Consequently, the change in surgeons, as well as their increasing experience over time, could be the reason why more SLNs per patient were detected when using the magnetometer technique. Routine histopathological examination of the dissected LNs and the distinct use of supplementary immunohistochemistry have also improved over the years at our hospital. Thus, the higher prevalence of LNI found in patients being treated with magnetometer-guided sPLND could have partly resulted from these developments. Furthermore, the advancements in PCa diagnostics as well as treatment options for low risk PCa (e.g., active surveillance) have led to updates in guidelines (e.g., EAU) for PCa over time with implications for clinical practice and a stage shift [46]. Therefore, it is unsurprising that the two patient cohorts in our retrospective study also differed in their clinical tumor properties. For example, the patient group treated with magnetometer-guided sPLND had more advanced tumors according to biopsy Gleason score (Table 1). As reflected in our results, the Gleason score may serve as a predictor for LNI [30,39], and this might partly explain the differences in the rates of LNI between the two sPLND techniques. In agreement with this, the observed differences in the rates of LNI are no longer present after propensity score analysis through optimal pair matching of the data according to clinical information. Nevertheless, the rates of LNI in both sPLND techniques accurately match the rates predicted by our sentinel-based nomogram [30]. Thus, our results show that both magnetometer- and radioisotope-guided sPLND techniques are reliable tools for LN staging in PCa. Since the performance of the magnetic method seems to be even slightly better and is also associated with a simpler clinical workflow, we will continue to use it to replace the radioactive sentinel procedure if there are no contraindications.

The magnetic SLN marking technique has some limitations [23,24,29]. The magnetic approach is not applicable in patients with pacemakers or other implanted electronic devices in the chest wall as well as those with hypersensitivity to iron or with iron overload disease. Furthermore, the detectability of the magnetic tracer is reduced in patients with metal implants such as hip prostheses or other metallic pelvic implants. In these cases, patients would still benefit from radioisotope-guided sPLND.

The sentinel approach bears one significant drawback. When LNs are fully metastasized, or lymphatic pathways are blocked, the afferent lymph could be directed to other LNs, which might not necessarily be SLNs [47,48]; this has been observed, for example, in inguinal SLNs in penile carcinoma [49,50]. Performing sPLND alone in these cases would yield false-negative results. The false-negative rates observed in our study were as low as those reported in other studies [19,22,35,48]. Thus, the diagnostic value of sPLND for LN staging in patients with PCa is similar to that when using a combination of sPLND and ePLND; this would have a higher prevalence of LNI than using ePLND alone [19,22,32,41].

In addition to the selective removal of LNs bearing the highest risk of containing metastases, the sentinel approach also enables surgical treatment, which is adjusted to the patient’s individual lymphatic drainage situation [19,25,48]. Consequently, we identified a considerable proportion of LN metastases outside the conventional ePLND template [33] using both sPLND techniques. These findings are well in line with those of other studies and highlight the diagnostic value of sPLND for LN staging in patients with PCa [19,22,25,36,51].

## 5. Conclusions

Our study confirms the diagnostic accuracy of magnetometer-guided sPLND in nodal staging in patients with PCa. We suggest magnetometer-guided sPLND as a reliable and promising alternative sPLND technique in PCa treatment to overcome the technical and logistic disadvantages of radioisotope marking. Furthermore, our results highlight the additional diagnostic value of the sentinel technique in PCa because it allows not only the selective removal of LNs bearing the highest risk of containing metastases but also surgical treatment that is adapted to the patient’s individual lymphatic drainage situation.

## Figures and Tables

**Figure 2 cancers-13-05821-f002:**
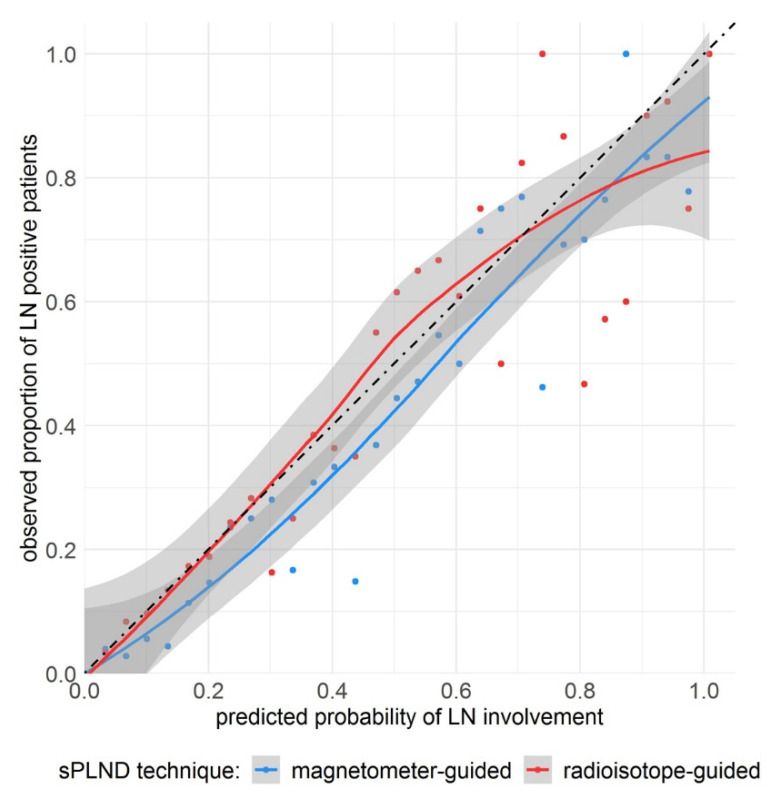
Observed proportion of patients with lymph node (LN) positivity in relation to the probability of LN involvement as predicted by the nomogram for magnetometer (blue)- and radioisotope (red)-guided sentinel pelvic lymph node dissection (sPLND). Gray shaded areas represent the 95% confidence intervals of the smoothed blue and red curves, respectively. The black dot-dashed line represents the ideal curve wherein the predicted probabilities and observed proportions match perfectly.

**Figure 3 cancers-13-05821-f003:**
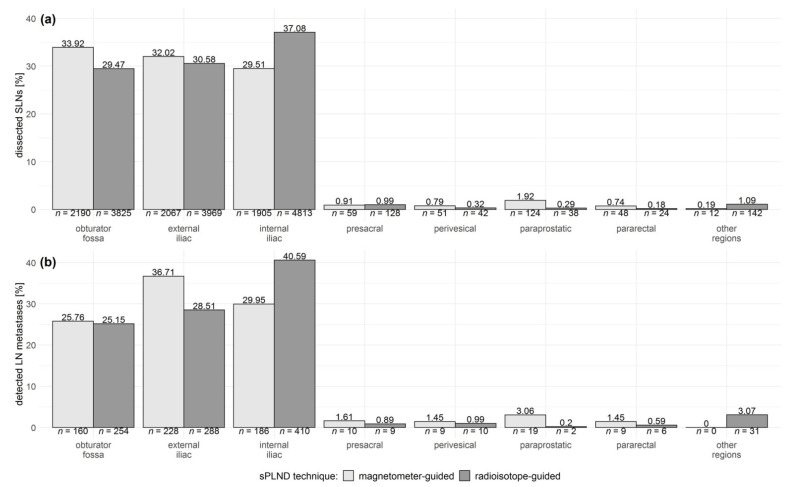
Anatomical distribution of (**a**) the dissected sentinel lymph nodes (SLNs) and (**b**) the lymph node (LN) metastases detected using magnetometer (light gray bars)- and radioisotope (dark gray bars)-guided sentinel pelvic lymph node dissection (sPLND) techniques, respectively. Data of the anatomic region were not available for 22 SLNs dissected using magnetometer-guided sPLND.

**Table 1 cancers-13-05821-t001:** Patient characteristics.

Method	Magnetometer-Guided sPLND			Radioisotope-Guided sPLND		
	Overall	pN0	pN1	Overall	pN0	pN1
*n* (%)	848	655 (77.24)	193 (22.76)	2092	1696 (81.07)	396 (18.93)
Age (IQR)	67 (62–71)	67 (61–71)	68 (64–73)	67 (62–71)	67 (61–71)	68 (63–71)
Total PSA ng/mL (IQR)	8.7 (6.1–13.5)	8 (5.8–11.8)	12.8 (8.6–27.7)	7.8 (5.5–12.5)	7.2 (5.3–10.9)	12.0 (7.9–20.6)
Dissected LNs (IQR)	14 (10–18)	13 (10–17)	16 (12–21)	10 (7–14)	10 (7–13)	12 (9–15)
Positive LNs (IQR)	0 (-)	0 (-)	2 (1–4)	0 (-)	0 (-)	2 (1–3)
Dissected SLNs (IQR)	7 (4–10)	7 (5–10)	6 (4–10)	6 (4–8)	6 (4–8)	6 (3–8)
Positive SLNs (IQR)	0 (-)	0 (-)	1 (1–2)	0 (-)	0 (-)	1 (1–2)
Clinical tumor stage (%)	*		*	**	**	**
cT1	436 (51.42)	397 (60.61)	39 (20.21)	1129 (53.97)	1027 (60.55)	102 (25.76)
cT2	368 (43.40)	244 (37.25)	124 (64.25)	919 (43.93)	658 (38.80)	261 (65.91)
cT3	41 (4.83)	14 (2.14)	27 (13.99)	36 (1.72)	6 (0.35)	30 (7.58)
cT4	2 (0.24)	0	2 (1.04)	2 (0.10)	0	2 (0.51)
Biopsy Gleason sum (%)				***	***	
≤6	162 (19.10)	150 (22.90)	12 (6.22)	998 (47.71)	938 (55.31)	60 (15.15)
=7 (3 + 4)	402 (47.41)	354 (54.05)	48 (24.87)	724 (34.61)	570 (33.61)	154 (38.89)
=7 (4 + 3)	129 (15.21)	88 (13.44)	41 (21.24)	191 (9.13)	109 (6.43)	82 (20.71)
≥8	155 (18.28)	63 (9.62)	92 (47.67)	176 (8.41)	76 (4.48)	100 (25.25)
Postoperative Gleason sum (%)						
≤6	30 (3.54)	30 (4.58)	0	349 (16.68)	348 (20.52)	1 (0.25)
=7 (3 + 4)	443 (52.24)	423 (64.58)	20 (10.36)	1122 (53.63)	1052 (62.03)	70 (17.68)
=7 (4 + 3)	216 (25.47)	147 (22.44)	69 (35.75)	420 (20.08)	230 (13.56)	190 (47.98)
≥8	159 (18.75)	55 (8.40)	104 (53.89)	201 (9.61)	66 (3.89)	135 (34.09)
Pathologic tumor stage (%)						
pT1c	2 (0.24)	2 (0.31)	0	1 (0.05)	1 (0.06)	0
pT2a	41 (4.83)	41 (6.26)	0	184 (8.80)	180 (10.61)	4 (1.01)
pT2b	21 (2.48)	19 (2.90)	2 (1.04)	40 (1.91)	39 (2.30)	1 (0.25)
pT2c	399 (47.05)	390 (59.54)	9 (4.66)	1086 (51.91)	1048 (61.79)	38 (9.60)
pT3a	178 (20.99)	141 (21.53)	37 (19.17)	407 (19.46)	300 (17.69)	107 (27.02)
pT3b	197 (23.23)	61 (9.31)	136 (70.47)	318 (15.20)	113 (6.66)	205 (51.77)
pT4	10 (1.18)	1 (0.15)	9 (4.66)	56 (2.68)	15 (0.88)	41 (10.35)

Data are presented as median (interquartile range) or frequency (percentage). sPLND: sentinel pelvic lymph node dissection; pN: pathologic nodal stage; IQR: interquartile range; (S)LN: (sentinel) lymph node; PSA: prostate-specific antigen; * clinical T-category could not be assessed in one patient (pN1); ** clinical T-category could not be assessed in six patients (pN0: *n* = 5, pN1: *n* = 1); *** incomplete biopsy data in three patients (pN0).

**Table 2 cancers-13-05821-t002:** Comparison between magnetometer-guided sPLND and radioisotope-guided sPLND either with original data or with propensity score-adjusted data.

Comparison	Original Results	Test Statistic	Adjusted Results	Test Statistic
SLN detection rate	98.11% (832) vs. 98.18% (2054)	χ^2^ < 0.001, df = 1, *p* = 1	98.11% (832) vs. 95.87% (812)	χ^2^ = 6.55, df = 1, *p* = 0.011 *
Number of dissected SLNs	7 (4–10) vs. 6 (4–8)	W = 1,059,411, *p* < 0.001 ***	7 (4–10) vs. 5 (3–7)	W = 471,031, *p* < 0.001 ***
Rate of LNI	22.76% (193) vs. 18.93% (396)	χ^2^ = 5.29, df = 1, *p* = 0.021 *	22.76% (192) vs. 25.97% (220)	χ^2^ = 2.34, df = 1, *p* = 0.126
Rate of LN+ but SLN-	7.25% (14) vs. 9.85% (39)	χ^2^ = 0.77, df = 1, *p* = 0.379	7.29% (14) vs. 14.55% (32)	χ^2^ = 4.73, df = 1, *p* = 0.030 *
False-negative rate	3.63% (7) vs. 5.05% (20)	χ^2^ = 0.32, df = 1, *p* = 0.572	3.65% (7) vs. 5.91% (13)	χ^2^ = 0.700, df = 1, *p* = 0.403

Data are presented as percentage (n) or median (interquartile range). (S)LN: (sentinel) lymph node; LNI: lymph node involvement; LN+: lymph node positivity; SLN-: sentinel lymph node negativity. * 5% significance level, *** 0.1% significance level.

## Data Availability

The data presented in this study are available upon reasonable request from the corresponding author.

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
