# Peer review of "Magnetometer-Guided Sentinel Lymph Node Dissection in Prostate Cancer: Rate of Lymph Node Involvement Compared with Radioisotope Marking"

_cancers, 2021, doi:10.3390/cancers13225821_

Round 1

Reviewer 1 Report

The manuscript is a retrospective study comparing the magnetic-guided and standard techniques in terms of their ability to detect lymph node metastases in prostate cancer. More sentinel lymph nodes were detected per patient with the magnetometer. 

The findings are interesting but the following need to be included:

  1. Authors compare two different techniques in different periods with no randomisation or matching. Together with the unadjusted results, propensity score matching needs to be performed and presented.
  2. The STROBE checklist needs to be followed
  3. The Discussion needs to take the propensity score analysis into consideration.

Reviewer 2 Report

In the current study, authors have presented an alternative modality of pelvic sentinel-lymphadenectomy (sPLND) in prostate cancer patients that is by magnetometer guidance rather than the conventional radioisotope-guided technique. The retrospective study presented a cohort of 848 and 2092 patient who underwent magnetometer-guided and radioisotope-guided sPLND, respectively. Authors compared the lymph node prostate cancer metastases discovery rate, and the anatomical distribution of the metastases discovery, and the data showed a very similar discovery rate comparing the two modalities. There are many advantages of carrying out magnetometer-guided sPLND as introduced. I believe the paper presented a great feasibility of the magnetometer-guided sPLND.

  1. Can authors show some representative IHC stainings of the discovered metastases between the two modalities from the study?

  1. As a retrospective study, it is important to show since the study, has there been any follow ups with the patients? How are the disease recurrence rate and disease free survival rate from the 2 groups of the patients?

Round 2

Reviewer 1 Report

The current version approves the manuscript

Author Response

We thank you again for your very helpful review and are pleased with your final approval for publication.
